# Nuclear magnetic resonance-based metabolomics with machine learning for predicting progression from prediabetes to diabetes

Jiang Li[1†], Yuefeng Yu[1†], Ying Sun[1], Yanqi Fu[1], Wenqi Shen[1], Lingli Cai[1], Xiao Tan[2,3], Yan Cai[4], Ningjian Wang[1], Yingli Lu[1*‡], Bin Wang[1*‡]

[1]Institute and Department of Endocrinology and Metabolism, Shanghai Ninth People's Hospital, Shanghai Jiao Tong University School of Medicine, Shanghai, China; [2]Department of Medical Sciences, Uppsala University, Uppsala, Sweden; [3]Department of Big Data in Health Science, School of Public Health, Zhejiang University School of Medicine, Hangzhou, China; [4]Department of Endocrinology, the Fifth Affiliated Hospital of Kunming Medical University, Yunnan Honghe Prefecture Central Hospital (Ge Jiu People's Hospital), Yunnan, China

*For correspondence:
luyingli2008@126.com (YL);
binwang1126@163.com (BW)

†These authors contributed equally to this work

Present address: ‡Institute and Department of Endocrinology and Metabolism, Shanghai Ninth People's Hospital, Shanghai Jiao Tong University School of Medicine, Shanghai, China

## Abstract

**Background:** Identification of individuals with prediabetes who are at high risk of developing diabetes allows for precise interventions. We aimed to determine the role of nuclear magnetic resonance (NMR)-based metabolomic signature in predicting the progression from prediabetes to diabetes.

**Methods:** This prospective study included 13,489 participants with prediabetes who had metabolomic data from the UK Biobank. Circulating metabolites were quantified via NMR spectroscopy. Cox proportional hazard (CPH) models were performed to estimate the associations between metabolites and diabetes risk. Supporting vector machine, random forest, and extreme gradient boosting were used to select the optimal metabolite panel for prediction. CPH and random survival forest (RSF) models were utilized to validate the predictive ability of the metabolites.

**Results:** During a median follow-up of 13.6 years, 2525 participants developed diabetes. After adjusting for covariates, 94 of 168 metabolites were associated with risk of progression to diabetes. A panel of nine metabolites, selected by all three machine-learning algorithms, was found to significantly improve diabetes risk prediction beyond conventional risk factors in the CPH model (area under the receiver-operating characteristic curve, 1 year: 0.823 for risk factors + metabolites vs 0.759 for risk factors, 5 years: 0.830 vs 0.798, 10 years: 0.801 vs 0.776, all p < 0.05). Similar results were observed from the RSF model. Categorization of participants according to the predicted value thresholds revealed distinct cumulative risk of diabetes.

**Conclusions:** Our study lends support for use of the metabolite markers to help determine individuals with prediabetes who are at high risk of progressing to diabetes and inform targeted and efficient interventions.

**Funding:** Shanghai Municipal Health Commission (2022XD017). Innovative Research Team of High-level Local Universities in Shanghai (SHSMU-ZDCX20212501). Shanghai Municipal Human Resources and Social Security Bureau (2020074). Clinical Research Plan of Shanghai Hospital Development Center (SHDC2020CR4006). Science and Technology Commission of Shanghai Municipality (22015810500).

## eLife assessment

This **important** study combines prospective cohort, metabolomics and machine learning to identify a panel of nine circulating metabolites that improved the ability in risk prediction of progression from prediabetes to diabetes. The findings are **convincing**, and using current state-of-the-art methods the data and analyses support the claims. This paper provides insights into the integration of these metabolites into clinical and public health practice.

## Introduction

Prediabetes, an intermediate stage of glucose dysregulation that blood glucose levels are elevated but lower than in diabetes, has become a burgeoning global health emergency (*Echouffo-Tcheugui and Selvin, 2021*). Prediabetes affected approximately 720 million individuals worldwide in 2021, with a project to 1 billion people by 2045 (*Sun et al., 2022*). Approximately 5–10% of people with prediabetes progress to having diabetes each year and the lifetime conversion rate to diabetes could be as high as 70% (*Tabák et al., 2012*; *Ligthart et al., 2016*). Therefore, preventing or delaying diabetes development among people with prediabetes will have substantial clinical and public health benefits.

Although lifestyle modification and medical therapy have been proven to be effective in preventing or delaying the diabetes onset among people with prediabetes (*Gong et al., 2019*; *DeFronzo et al., 2011*; *Herman, 2023*), the substantial cost of modification programs and medications as well as drug-related side effects limit the widespread delivery of such interventions in this large high-risk population (*Roberts et al., 2017*; *Piller, 2019*). Notably, the progression from prediabetes to diabetes is highly heterogeneous, and a fraction of individuals with prediabetes may regress to normoglycemia without treatment (*Shang et al., 2019*). Therefore, identifying targeted population who are at high risk of developing diabetes is the key step to tailor precise and efficient interventions. Glycemic indicators alone for risk stratification are deficient, with fasting glucose and glycosylated hemoglobin A1c (HbA1c) being convenient but less sensitive, while post-load glucose tolerance being sensitive but unfeasible in practice on a large scale (*Phillips et al., 2014*; *Ferrannini, 2014*). In addition, several risk assessment models based on conventional clinical variables have been developed, but most of which had comparatively low performance and failed to take follow-up time into account (*Yokota et al., 2017*; *Cahn et al., 2020*; *Liang et al., 2021*).

Plasma metabolomics using high-throughput techniques could provide a comprehensive profiling of small-molecule metabolites in a specific physiological period, which might yield valuable information for risk prediction. Previous studies have implied that incorporating circulating metabolites into basic models with conventional risk factors could improve prediction of diabetes risk (*Merino et al., 2018*; *Peddinti et al., 2017*; *Rebholz et al., 2018*). However, we are aware of only one study that has assessed the relationship between metabolomic profiling and the progression to diabetes among individuals with prediabetes and investigated the predictive values of metabolites (*Ren et al., 2021*). Nevertheless, it was limited by a nested case–control study design with a relatively short follow-up (median 5 years) and small sample size (*n* = ~300). Whether addition of metabolic biomarkers improves the ability in predicting the progression from prediabetes to diabetes in prospective settings remains largely unknown.

To address these knowledge gaps, in the current study, we aimed to examine the longitudinal associations of circulating metabolic biomarkers, quantified using high-throughput nuclear magnetic resonance (NMR), with the risk of incident diabetes among individuals with prediabetes from the UK Biobank. Moreover, we evaluated whether metabolic signature adds anything to prediction models for diabetes development and risk stratification.

## Methods

### Study design and participants

The UK Biobank is a large population-based prospective cohort study enrolling more than 500,000 community-dwelling adults from 22 assessment centers across the UK between 2006 and 2010 (*Sudlow et al., 2015*; *Allen et al., 2012*). Participants completed touchscreen questionnaires and physical measurements and provided blood samples at baseline. The study was approved by the

Northwest Multicenter Research Ethics Committee (REC reference for UK Biobank 11/NW/0382), and all participants provided informed consent.

For the identification of metabolomic biomarkers associated with the progression from prediabetes to diabetes, the current study focused on participants with prediabetes at baseline with available circulating metabolite data. The diagnosis of prediabetes was defined by an HbA1c level of 5.7–6.4% (39–47 mmol/mol) in participants without diabetes, according to the American Diabetes Association (ADA) criteria (**ElSayed et al., 2023**). After excluding individuals who developed diabetes or died within 1 month from the baseline, 13,489 participants with prediabetes were included in the final analyses.

## Metabolite quantification

The metabolomics analysis of approximately 118,000 non-fasting ethylenediaminetetraacetic acid (EDTA) plasma samples at baseline was performed using the high-throughput NMR platform in Nightingale Health's laboratories of Finland. Details of the metabolic profiling platform and experimentation have been described elsewhere (**Würtz et al., 2017**; **Soininen et al., 2015**; **Zhang et al., 2022**). In brief, the EDTA samples were collected and stored at −80°C. Before preparation, frozen samples were slowly thawed at +4°C overnight and were centrifuged (3400 × $g$) for 3 min. Each sample was analyzed with a spectrometer and the metabolic biomarkers were quantified using Nightingale Health's proprietary software. The quality control procedures were implemented during the whole process and only samples and biomarkers that underwent the quality control process were stored in the UK Biobank dataset.

The metabolic biomarker profiling by Nightingale Health's NMR platform provides consistent results over time and across spectrometers. Furthermore, the sample preparation is minimal in the Nightingale Health's metabolic biomarker platform, circumventing all extraction steps. These aspects result in highly repeatable biomarker measurements. Pre-specified quality metrics were agreed between UK Biobank and Nightingale Health to ensure consistent results across the samples, and pilot measurements were conducted. Nightingale Health performed real-time monitoring of the measurement consistency within and between spectrometers throughout the UK Biobank samples. Two control samples provided by Nightingale Health were included in each 96-well plate for tracking the consistency across multiple spectrometers. Furthermore, two blind duplicate samples provided by the UK Biobank were included in each well plate, with the position information unlocked only after results delivery. Coefficient of variation (CV) targets across the metabolic biomarker profile were pre-specified for both Nightingale Health's internal control samples and UK Biobank's blind duplicates. The targets were met for each consecutively measured batch of ~25,000 samples. For the majority of the metabolic biomarkers, the CVs were below 5% (https://biobank.ndph.ox.ac.uk/showcase/refer.cgi?id=3000). Furthermore, the distributions of measured biomarkers from five sample batches indicated absence of batch effects (https://biobank.ctsu.ox.ac.uk/ukb/ukb/docs/nmrm_app1).

A total of 249 metabolic biomarkers (168 directly measured and 81 ratios of these), spanning lipids, lipoprotein subclass, fatty acids, amino acids, ketone bodies, and glycolysis metabolites were quantified for each sample. In the present study, we analyzed 168 metabolic biomarkers that were directly measured (**Supplementary file 1**). The values of all metabolites were transformed using natural logarithmic transformation (ln[$x$ + 1]) followed by $Z$-transformation.

## Covariate collection

Information on covariates was collected through a self-completed touchscreen questionnaire or verbal interview, including age, sex, ethnicity (White people or others), Townsend Deprivation Index, household income (high: ≥£52,000, medium: £18,000–£51,999, and low: <£18,000), education (college/university degree or others), employment status (current working, retired, or other), smoking status (never, previous, or current smoking), moderate alcohol (alcohol intake >0 g and ≤14 g/day for women and alcohol intake >0 g and ≤28 g/day for men), physical activity, healthy diet score, healthy sleep score, family history of diabetes (yes or no), history of cardiovascular disease (CVD, yes or no), history of hypertension (yes or no), history of dyslipidemia (yes or no), history of chronic lung diseases (CLD, e.g. chronic bronchitis, emphysema, and chronic obstructive pulmonary disease, yes or no), and history of cancer (yes or no). The Townsend Deprivation Index is a composite measure of area-level socioeconomic deprivation, with a higher score indicating higher levels of socioeconomic deprivation. Physical

activity was measured by the metabolic equivalent task (MET) (sum of days performing walking, moderate activity, and vigorous activity) (*Liang et al., 2023*). A healthy diet score was calculated based on the intake of vegetables (≥median), fruits (≥median), fish (≥median), red meat (<median), and processed red meat (<median) (*Wang et al., 2023*). One point was given for each favorable diet factor and the total diet score ranges from 0 to 5. A healthy sleep score was evaluated based on insomnia (sometimes or never), sleep duration (7–8 hr), chronotype (morning person), daytime sleepiness (sometimes or never), and snoring (no) (*Song et al., 2023*). Each favorable sleep factor was given a score of 1, with the total sleep score ranging from 0 to 5. The history of dyslipidemia incorporated information on both the medical history of dyslipidemia and the use of lipid-lowering medications.

Physical measurements including blood pressure, height, weight, waist circumference (WC), and hip circumference (HC) were measured using calibrated instruments with standard protocols by trained nurses. Blood pressure was measured using the Omron automatic digital monitor and two measurements were obtained at a few minutes' intervals. We calculated the mean systolic (SBP) and diastolic blood pressure (DBP) from two measurements. Body mass index (BMI) was calculated as weight in kilograms divided by the square of height in meters (kg/m²). The HbA1c level was measured by high-performance liquid chromatography with the VARIANT II Turbo analyzer (Bio-Rad Laboratories). Missing covariates were imputed by the median value for continuous variables and a missing indicator for categorical variables.

## Ascertainment of diabetes

Incident diabetes was ascertained from hospital inpatient records, death registers, and primary care records, according to the International Classification of Diseases, 10th revision (ICD-10) codes. Detailed information about the linkage procedure is available from https://content.digital.nhs.uk/services. The follow-up time was calculated from the baseline to the occurrence of diabetes, death, or the censoring date (March 30, 2023), whichever came first.

## Statistical analyses

Baseline characteristics were presented as numbers (percentages) for categorical variables and means (standard deviations, SDs) for continuous variables, respectively. Continuous variables were assessed for statistical differences using *t*-test and categorical variables were evaluated using the $\chi^2$ test. Overall schematic workflow of the study is shown in *Figure 1*.

## Metabolite selection

We first used Cox proportional hazard (CPH) model to assess the associations between individual metabolites and risk of diabetes progression with adjustment for sociodemographic covariates (age, sex, ethnicity, education, Townsend Deprivation Index, employment status, and household income), family history of diabetes, health conditions (history of CVD, hypertension, dyslipidemia, CLD, and cancer), physical measurements (BMI, WC, HC, SBP, and DBP), lifestyle factors (smoking status, moderate alcohol, healthy diet score, healthy sleep score, and physical activity), and HbA1c. The potential confounders were selected based on prior knowledge of the risk factors for diabetes. Metabolites that were significantly associated with incident diabetes (p < 0.05/168) were retained.

Second, we performed priority-Lasso to deal with multicollinearity in high dimensional data and to retain variables with nonzero coefficients. Priority-Lasso is a Lasso-based intuitive analysis strategy, which uses prior knowledge regarding the outcome by defining the blocks of different types of predictor variables (*Klau et al., 2018*). In this study, we defined the 24 covariates as block 1, while all metabolites significantly associated with diabetes risk in the CPH model were defined as block 2. The penalization parameter $\lambda$ was determined as values with maximum partial-likelihood in a 10-fold cross-validation.

Third, three machine-learning models including supporting vector machine, random forest, and extreme gradient boosting were adopted to further evaluate the importance of the Lasso-selected metabolites, as they can model nonlinear and nonadditive relations more flexibly (*Morgenstern et al., 2021*). Models were built by 10-fold cross-validation through the 'caret' package. Common signals detected across diverse approaches are more likely to represent the strongest and true patterns in the data. We chose the intersection set of the top 20 most important variables selected by the three

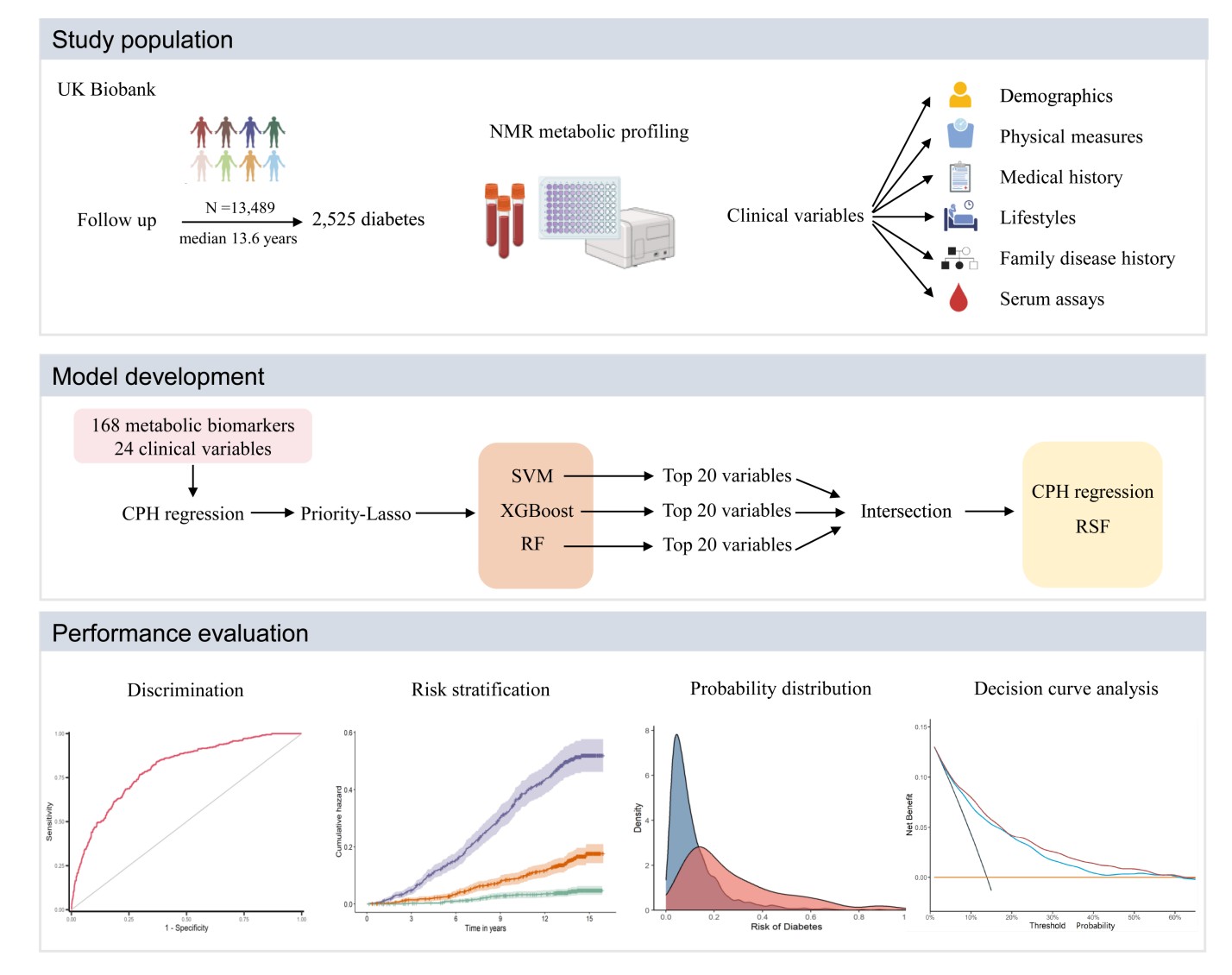

**Figure 1.** Overall schematic workflow of the study. CPH, Cox proportional hazard; NMR, nuclear magnetic resonance; RF, random forest; RSF, random survival forest; SVM, supporting vector machine; XGBoost, extreme gradient boosting.

machine-learning models, after balancing the performance of the final diabetes risk prediction model and the clinical applicability associated with measurement costs of metabolites.

## Model development

Participants were randomly subclassified into a training set and a test set at a ratio of 8:2 and two common algorithms for survival data including CPH model and random survival forest (RSF) (*Qiu et al., 2020*) were adopted for model development. RSF, as a machine-learning method, is designed to be used specifically for survival outcome prediction and has shown promising results in various settings (*Rahman et al., 2023*; *Kwak et al., 2021*). It builds many decision trees using split points based on the log-rank test to identify different survival statuses and produces the predicted probability for an individual derived from the average prediction across all trees (*Ishwaran et al., 2008*). The RSF model was fitted using the 'randomForestSRC' package and the grid search method was used for hyperparameter tuning (number of trees, number of variables to possibly split at each node, and minimum size of terminal node). Specifically, the grid search method was used to tune hyperparameters among the RSF model, through minimizing out-of-sample or out-of-bag error (*Janitza and Hornung, 2018*). Each tree in the RSF is constructed from a random sample of the data, typically a bootstrap sample

or 63.2% of the sample size (as in the present study). Consequently, not all observations are used to construct each tree. The observations that are not used in the construction of a tree are referred to as out-of-bag observations. In an RSF model, each tree is built from a different sample of the original data, so each observation is 'out-of-bag' for some of the trees. The prediction for an observation can then be obtained using only those trees for which the observation was not used for the construction. A classification for each observation is obtained in this way and the error rate can be estimated from these predictions. The resulting error rate is referred to as the out-of-bag error. Through calculating the out-of-bag error in each iteration, the best hyperparameters were finally determined. The hyperparameters to be tuned and range of grid search in the present study were below: number of trees (50–1000, by 50), number of variables to possibly split at each node (3–6, by 1), and minimum size of terminal node (1–20, by 1) (*Tian et al., 2023*).

## Model evaluation

The model performance was assessed in the test set. The time-dependent area under the receiver-operating characteristic curve (AUROC) was used to evaluate the model's discrimination ability. Continuous net reclassification improvement (NRI) and absolute integrated discrimination improvement (IDI) were used to assess whether adding the selected metabolites could improve risk discrimination and reclassification for the risk of progression from prediabetes to diabetes over the basic model that was built on 10 conventional clinical variables (age, sex, Townsend Deprivation Index, family history of diabetes mellitus, BMI, WC, HC, SBP, DBP, and HbA1c) (*Wilson et al., 2007*). The calibration ability of the model was estimated using calibration curve. Furthermore, we used decision curve analysis (DCA) to assess the clinical usefulness of prediction model-based guidance for prediabetes management, which calculates a clinical 'net benefit' for one or more prediction models in comparison to default strategies of treating all or no patients (*Vickers et al., 2019*). To facilitate risk stratification, we classified participants into two risk groups according to the predictive value using 'surv_cutpoint' function in the 'survminer' R package (*Fan et al., 2023*). We also divided participants into three categories according to the tertiles of probability. In addition, we included 90,688 participants with normal glucose from the UK Biobank and divided them into the training and test sets using an 8:2 ratio to further investigate the additive value of the selected metabolites in diabetes prediction among participants with normoglycemia. All analyses were conducted in R software (version 4.2.2). A two-sided p value <0.05 was considered statistically significant. To control for the false discovery rate in the association between multiple metabolic biomarkers and incident diabetes, Bonferroni correction for p value (p < 0.05/168) was used.

## Results

### Baseline characteristics

Among the 13,489 participants with baseline prediabetes, the mean age was 59.6 (SD, 7.1) years, and 6166 (45.7%) were males. During a median follow-up of 13.6 (12.3–14.6) years, 2525 (18.7%) participants progressed to diabetes. Baseline characteristics of the study population stratified by incident diabetes are summarized in *Table 1*. Participants who developed diabetes were more likely to be male, non-White, less educated, more deprived, and smokers. They also tended to have a family history of diabetes, comorbidities such as CVD, hypertension, dyslipidemia, and CLD, and higher levels of BMI, WC, and HC.

### Identification of metabolic biomarkers for progression to diabetes

After adjusting for covariates and correcting for multiple testing, 94 of 168 metabolic biomarkers were significantly associated with the risk of incident diabetes (*Figure 2* and *Supplementary file 2*). Concentrations of very-low-density lipoprotein (VLDL) particles, particularly larger VLDL particles and composition within larger VLDL, were strongly associated with progression to diabetes. Triglyceride in all lipoprotein subclasses also demonstrated strong positive associations with diabetes risk. In contrast, concentrations of larger high-density lipoprotein (HDL) particles and composition within these particles were inversely associated with incident diabetes. For lipoprotein particle diameter, larger HDL and low-density lipoprotein (LDL) particle sizes were associated with a lower risk of progression to diabetes, while larger VLDL particle size was associated with a higher risk.

**Table 1.** Baseline characteristics of participants with prediabetes stratified by incident diabetes status.

| Characteristics | Overall (*n* = 13489) | Diabetes (*n* = 2525) | Non-diabetes (*n* = 10964) | p value |
|---|---|---|---|---|
| Age, years | 59.6 (7.1) | 59.7 (7.1) | 59.6 (7.0) | 0.347 |
| Male | 6166 (45.7) | 1407 (55.7) | 4759 (43.4) | <0.001 |
| Education | | | | <0.001 |
| College or university | 3409 (25.3) | 498 (19.7) | 2911 (26.6) | |
| Others | 10056 (74.5) | 2022 (80.1) | 8034 (73.3) | |
| Unknown | 24 (0.2) | 5 (0.2) | 19 (0.2) | |
| Ethnicity | | | | 0.013 |
| White | 12172 (90.2) | 2239 (88.7) | 9933 (90.6) | |
| Others | 1293 (9.6) | 281 (11.1) | 1012 (9.2) | |
| Unknown | 24 (0.2) | 5 (0.2) | 19 (0.2) | |
| Employment status | | | | <0.001 |
| Working | 6608 (49.0) | 1172 (46.4) | 5436 (49.6) | |
| Retired | 5931 (44.0) | 1114 (44.1) | 4817 (43.9) | |
| Other | 787 (5.8) | 212 (8.4) | 575 (5.2) | |
| Unknown | 163 (1.2) | 27 (1.1) | 136 (1.2) | |
| Household income | | | | <0.001 |
| Low | 3529 (26.2) | 2734 (24.9) | 795 (31.5) | |
| Medium | 5659 (42.0) | 4666 (42.6) | 993 (39.3) | |
| High | 1897 (14.1) | 1611 (14.7) | 286 (11.3) | |
| Unknown | 2404 (17.8) | 1953 (17.8) | 451 (17.9) | |
| Townsend Deprivation Index | −1.0 (3.3) | −0.7 (3.4) | −1.1 (3.2) | <0.001 |
| Family history of DM | 3068 (22.7) | 786 (31.1) | 2282 (20.8) | <0.001 |
| History of CVD | 1392 (10.3) | 413 (16.4) | 979 (8.9) | <0.001 |
| History of hypertension | 4217 (31.3) | 985 (39.0) | 3232 (29.5) | <0.001 |
| History of dyslipidemia | 1932 (14.3) | 417 (16.5) | 1515 (13.8) | 0.001 |
| History of CLD | 1847 (13.7) | 413 (16.4) | 1434 (13.1) | <0.001 |
| History of cancer | | | | 0.056 |
| Yes | 1315 (9.7) | 215 (8.5) | 1100 (10.0) | |
| No | 12171 (90.2) | 2309 (91.4) | 9862 (89.9) | |
| Unknown | 3 (0.0) | 1 (0.0) | 2 (0.0) | |
| BMI, kg/m$^2$ | 29.0 (5.2) | 31.3 (5.3) | 28.4 (5.0) | <0.001 |
| WC, cm | 94.6 (13.5) | 101.3 (13.1) | 93.1 (13.1) | <0.001 |
| HC, cm | 105.4 (10.0) | 108.6 (10.8) | 104.6 (9.7) | <0.001 |
| Smoking status, % | | | | <0.001 |
| Never | 6478 (48.0) | 1104 (43.7) | 5374 (49.0) | |
| Previous | 4843 (35.9) | 1003 (39.7) | 3840 (35.0) | |
| Current | 2074 (15.4) | 397 (15.7) | 1677 (15.3) | |
| Unknown | 94 (0.7) | 21 (0.8) | 73 (0.7) | |
| Moderate alcohol | | | | 0.081 |

*Table 1 continued on next page*

*Table 1 continued*

| Characteristics | Overall (*n* = 13489) | Diabetes (*n* = 2525) | Non-diabetes (*n* = 10964) | p value |
|---|---|---|---|---|
| Yes | 3888 (28.8) | 689 (27.3) | 3199 (29.2) | |
| No | 9595 (71.1) | 1836 (72.7) | 7759 (70.8) | |
| Unknown | 6 (0.0) | 0 (0.0) | 6 (0.1) | |
| Healthy diet score | 3.3 (1.1) | 3.2 (1.1) | 3.3 (1.1) | <0.001 |
| Healthy sleep score | 3.5 (1.0) | 3.3 (1.1) | 3.6 (1.0) | <0.001 |
| Physical activity, METs | 10.4 (4.9) | 9.7 (5.1) | 10.6 (4.9) | <0.001 |
| SBP, mmHg | 141.3 (18.5) | 143.5 (18.2) | 140.8 (18.5) | <0.001 |
| DBP, mmHg | 83.3 (10.2) | 84.6 (10.4) | 83.0 (10.1) | <0.001 |
| HbA1c, % | 5.9 (0.2) | 6.0 (0.2) | 5.9 (0.2) | <0.001 |

Data were presented as means (standard deviations, SDs) for continuous variables and numbers (percentages) for categorical variables.
BMI, body mass index; DM, diabetes mellitus; CVD, cardiovascular disease; CLD, chronic lung disease; DBP, diastolic blood pressure; HbA1c, glycated hemoglobin A1c; HC, hip circumference; MET, metabolic equivalent of task; SBP, systolic blood pressure; WC, waist circumference.

Monounsaturated fatty acids and saturated fatty acids were positively associated with the risk of diabetes, whereas docosahexaenoic acid and the degree of fatty acid unsaturation were negatively associated with diabetes. Among the amino acids, higher concentrations of alanine, tyrosine, and branched-chain amino acid (BCAA) such as leucine and valine were associated with an increased risk of diabetes, but glutamine and glycine were inversely associated with diabetes. Neither of the ketone bodies showed an association with the risk of diabetes.

Of the 94 metabolites that were significantly associated with diabetes, 17 metabolites were selected by priority-Lasso (*Supplementary file 3*). When further evaluating the importance of these metabolites after adjustment for covariates using three machine-learning algorithms, the intersection of the top 20 important predictors identified a total of 9 metabolites, namely cholesteryl esters in large HDL, cholesteryl esters in medium VLDL, triglycerides in very large VLDL, average diameter for LDL particles, triglycerides in intermediate-density lipoprotein (IDL), glycine, tyrosine, glucose, and docosahexaenoic acid (*Figure 3* and *Supplementary file 4*).

## Model development and evaluation

Build upon the selected 9 metabolites and 10 clinical variables, there was no obvious difference in the AUROC obtained from CPH model (1 year: 0.823 [95% confidence interval, CI 0.702, 0.945]; 5 years: 0.830 [0.797, 0.864]; 10 years: 0.801 [0.778, 0.825]) and RSF model (1 year: 0.828 [0.723, 0.933]; 5 years: 0.820 [0.785, 0.855]; 10 years: 0.802 [0.778, 0.826]). Hence, we chose CPH model as the final model because of its simplicity and interpretability. The addition of selected metabolites consecutively outperformed the basic model with conventional clinical variables in diabetes risk prediction from 1 to 10 years (*Figure 4*). Specifically, the AUROC increased from 0.759 (95% CI 0.608, 0.911)–0.823 (0.702, 0.945), 0.798 (0.762, 0.834)–0.830 (0.797, 0.864), and 0.776 (0.750, 0.801) to 0.801 (0.778, 0.825) for 1-, 5-, and 10-year diabetes risk, respectively (*Table 2* and *Figure 5*). Results from continuous NRI and absolute IDI also demonstrated improvement in the risk prediction for progression to diabetes (*Table 2*), although the model calibration was not significantly improved (*Figure 6*). The decision curve analysis showed that the inclusion of the metabolites had a higher net benefit across the threshold probabilities of 0–0.35 for predicting 5-year diabetes risk and 0–0.55 for predicting 10-year diabetes risk (*Figure 7*).

We further categorized the participants from the test set into low- and high-risk groups according to the optimal threshold of the predicted value (1.02) reflecting the best risk difference. Compared with the low-risk group, participants in the high-risk group had a significantly higher cumulative risk of incident diabetes (log-rank p < 0.0001) (*Figure 8*). When participants were alternatively classified into low-, medium-, and high-risk groups according to the tertile cut-off point of the predicted value, the high-risk group showed the highest risk of developing diabetes, followed by the medium- and low-risk groups (log-rank p < 0.0001). Similar results were also observed when considering the competing risk

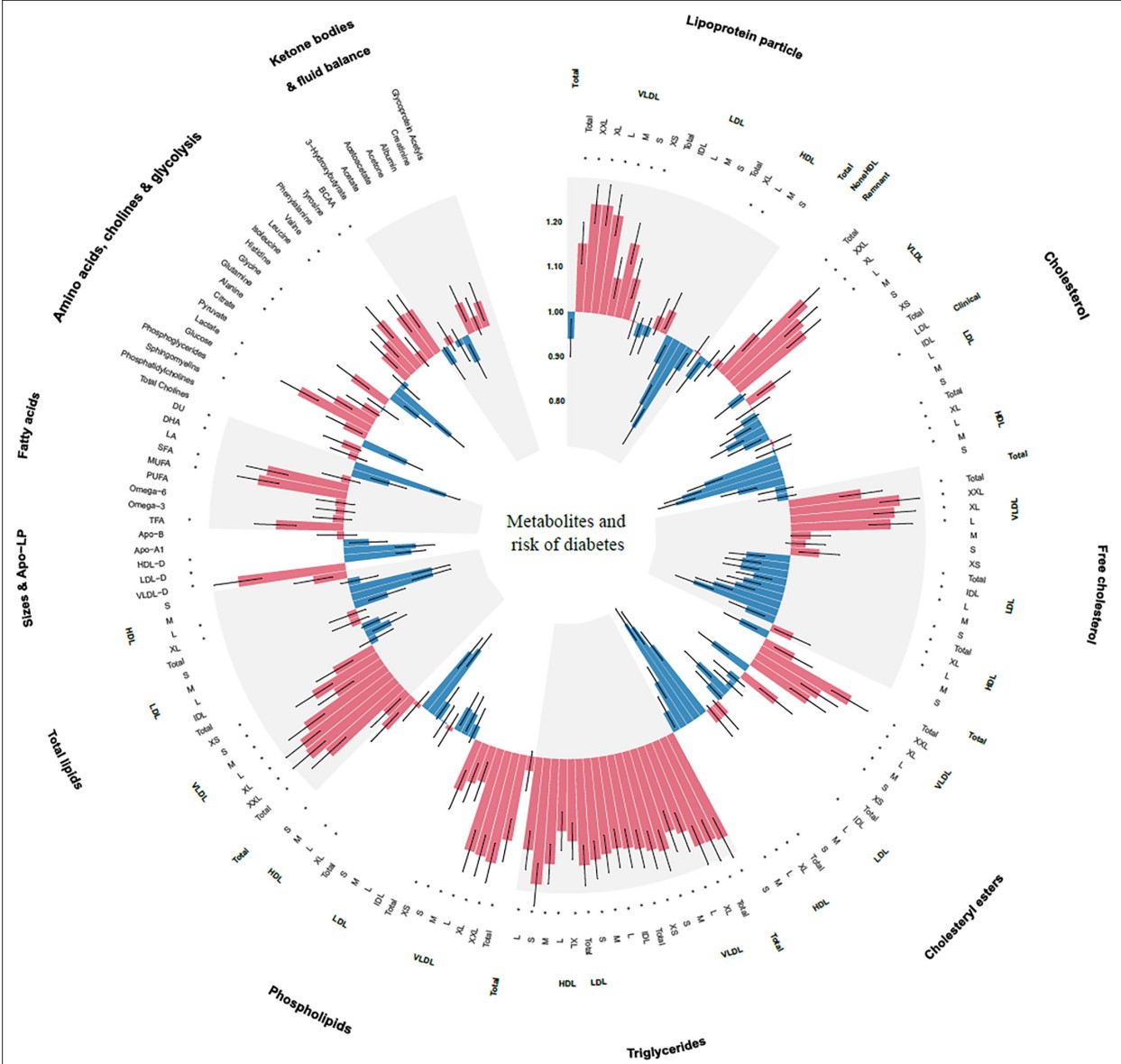

**Figure 2.** Associations of 168 metabolic biomarkers with risk of diabetes among 13,489 participants with prediabetes. Hazard ratios (HRs) were presented per 1 standard deviation (SD) higher of metabolic biomarkers on the natural log scale and were adjusted for age, sex, ethnicity, education, Townsend Deprivation Index, employment status, household income, family history of diabetes, history of CVD, history of hypertension, history of dyslipidemia, history of CLD, history of cancer, body mass index, waist circumference, hip circumference, smoking status, moderate alcohol, healthy diet score, healthy sleep score, physical activity, systolic blood pressure, diastolic blood pressure, and glycated hemoglobin A1c. *False discovery rate controlled p < 0.05/168. Apo-A1, apolipoprotein A1; Apo-B, apolipoprotein B; Apo-LP, apolipoprotein; BCAA, branched-chain amino acid; BMI, body mass index; CVD, cardiovascular disease; CLD, chronic lung disease; DHA, docosahexaenoic acid; FA, fatty acids; HDL, high-density lipoproteins; HDL-D, high-density lipoprotein particle diameter; IDL, intermediate-density lipoproteins; L, large; LA, linoleic acid; LDL, low-density lipoproteins; LDL-D, low-density lipoprotein particle diameter; LP, lipoprotein; M, medium; MUFA, monounsaturated fatty acids; PUFA, polyunsaturated fatty acids; S, small; SFA, saturated fatty acids; VLDL, very-low-density lipoproteins; VLDL-D, very-low-density lipoprotein particle diameter; XL, very large; XS, very small; XXL, extremely large.

from death (Fine–Gray p < 0.0001) (*Figure 8—figure supplement 1*). In addition, the predicted risk of diabetes within 1 year (p = 0.001), 5 years (p < 0.001), or 10 years (p <0.001) was generally higher among participants who progressed to diabetes than those who did not (*Figure 9*).

Among participants with normoglycemia, we also observed a significant improvement in the prediction of diabetes after the addition of metabolic biomarkers to the basic model. The AUROC increased from 0.821 (95% CI 0.736, 0.907) to 0.868 (0.802, 0.934), 0.790 (0.738, 0.842) to 0.811

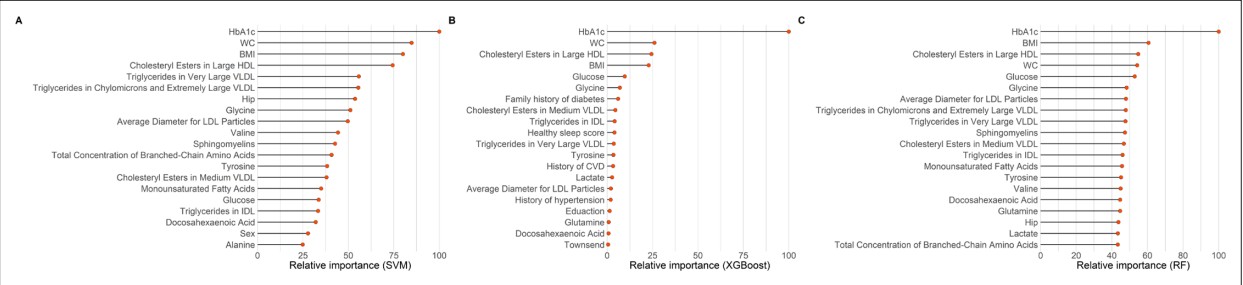

**Figure 3.** The top 20 important variables selected by three machine-learning models: (**A**) supporting vector machine (SVM); (**B**) extreme gradient boosting (XGBoost); (**C**) random forest (RF). The models were adjusted for age, sex, ethnicity, education, Townsend Deprivation Index, employment status, household income, family history of diabetes, history of CVD, history of hypertension, history of dyslipidemia, history of CLD, history of cancer, body mass index, waist circumference, hip circumference, smoking status, moderate alcohol, healthy diet score, healthy sleep score, physical activity, systolic blood pressure, diastolic blood pressure, and glycated hemoglobin A1c. CVD, cardiovascular disease; CLD, chronic lung disease; HDL, high-density lipoproteins; IDL, intermediate-density lipoproteins; LDL, low-density lipoproteins; VLDL, very-low-density lipoproteins.

(0.762, 0.860), and 0.791 (0.765, 0.816) to 0.806 (0.781, 0.831) for 1-, 5-, and 10-year diabetes risk, respectively (*Supplementary file 5*). The increases in NRI and IDI were similar to or slightly lower than those found among participants with prediabetes.

## Discussion

By leveraging data from the large UK Biobank cohort, this prospective study provided a comprehensive analysis of the associations of circulating metabolites with the risk of progression to diabetes and predictive ability in participants with prediabetes. We found that lipoprotein particles, lipoprotein particle size and composition, fatty acids, and amino acids were associated with the risk of incident diabetes. More importantly, our findings suggested that adding the selected metabolites (i.e., cholesteryl esters in large HDL, cholesteryl esters in medium VLDL, triglycerides in very large VLDL, average diameter for LDL particles, triglycerides in IDL, glycine, tyrosine, glucose, and docosahexaenoic acid) could significantly improve the risk prediction of progression from prediabetes to diabetes beyond the conventional clinical variables.

In the present study, the association between diabetes risk and lipid and lipoprotein profile, including VLDL particles and composition with larger VLDL, HDL particles and composition within

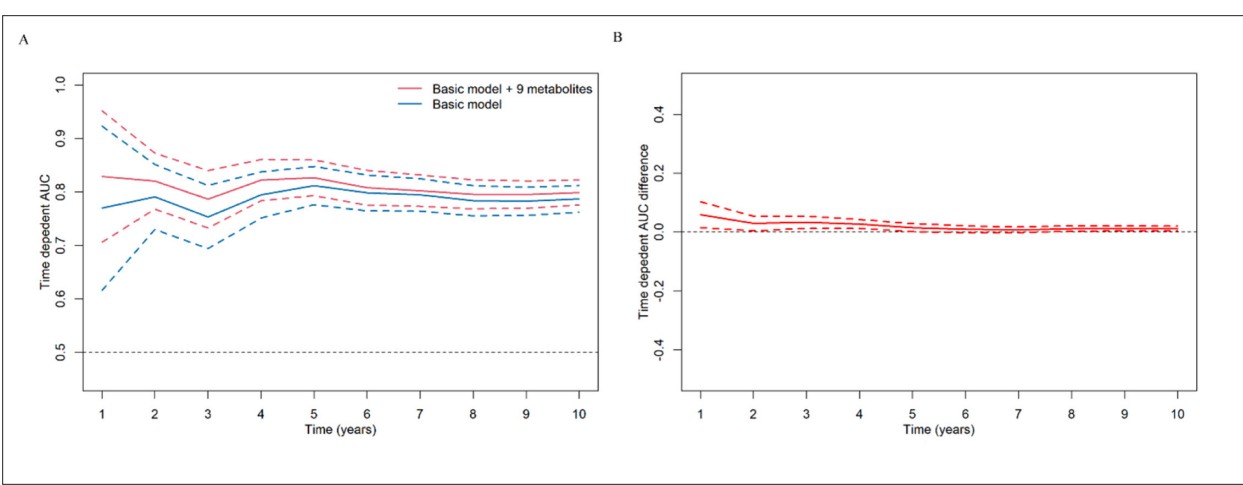

**Figure 4.** Consecutive area under time-dependent receiver-operating characteristic (AUROC) of basic model and basic model plus nine metabolites (**A**), and the difference of these two time-dependent AUROCs over time (**B**). The basic model used conventional clinical variables including age, sex, Townsend Deprivation Index, family history of diabetes mellitus, body mass index, waist circumference, hip circumference, systolic blood pressure, diastolic blood pressure, and glycated hemoglobin A1c. The selected nine metabolites included cholesteryl esters in large HDL, triglycerides in very large VLDL, glycine, average diameter for LDL particles, tyrosine, cholesteryl esters in medium VLDL, glucose, triglycerides in IDL, and docosahexaenoic acid. HDL, high-density lipoprotein; IDL, intermediate-density lipoprotein; LDL, low-density lipoprotein; VLDL, very-low-density lipoprotein.

**Table 2.** Performance of Cox proportional hazards regression models in prediction of the progression of prediabetes to diabetes.

| Performance metric | Basic model* | Basic model + nine metabolites† | p value |
|---|---|---|---|
| AUROC | | | |
| T = 1 year | 0.759 (0.608, 0.911) | 0.823 (0.702, 0.945) | 0.009 |
| T = 5 years | 0.798 (0.762, 0.834) | 0.830 (0.797, 0.864) | <0.001 |
| T = 10 years | 0.776 (0.750, 0.801) | 0.801 (0.778, 0.825) | <0.001 |
| Continuous NRI | | | |
| T = 1 year | Reference | 0.461 (0.134, 0.660) | <0.001 |
| T = 5 years | Reference | 0.400 (0.277, 0.483) | <0.001 |
| T = 10 years | Reference | 0.329 (0.252, 0.405) | <0.001 |
| Absolute IDI | | | |
| T = 1 year | Reference | 0.006 (−0.002, 0.020) | 0.132 |
| T = 5 years | Reference | 0.028 (0.017, 0.040) | <0.001 |
| T = 10 years | Reference | 0.040 (0.027, 0.054) | <0.001 |

AUROC, area under the receiver-operating characteristic curve; HDL, high-density lipoprotein; IDL, intermediate-density lipoprotein; IDI, absolute integrated discrimination improvement; LDL, low-density lipoprotein; NRI, net reclassification improvement; VLDL, very-low-density lipoprotein.

*Basic model: age, sex, Townsend Deprivation Index, family history of diabetes mellitus, body mass index, waist circumference, hip circumference, systolic blood pressure, diastolic blood pressure, and glycated hemoglobin A1c.

†The selected nine metabolic biomarkers: cholesteryl esters in large HDL, triglycerides in very large VLDL, glycine, average diameter for LDL particles, tyrosine, cholesteryl esters in medium VLDL, glucose, triglycerides in IDL, docosahexaenoic acid.

larger HDL, triglyceride, smaller HDL and LDL particle sizes, and larger VLDL particle sizes, were broadly consistent with previous studies in the general population (*Bragg et al., 2022c*; *Mackey et al., 2015*; *Bragg et al., 2022b*; *Bragg et al., 2022a*). BCAAs have been widely reported to be involved in the pathogenesis of diabetes, which might impair insulin signaling and lead to increased insulin secretion and pancreatic β-cell exhaustion (*Morze et al., 2022*). Furthermore, genetic association studies have shown higher BCAAs resulting from insulin resistance, which may in turn cause diabetes (*Lotta et al., 2016*; *Mahendran et al., 2017*). Our study confirmed the vital role of these metabolites in the progression to diabetes among individuals with prediabetes.

Several risk assessment models for predicting the risk of progression from prediabetes to diabetes have been reported (*Yokota et al., 2017*; *Cahn et al., 2020*; *Liang et al., 2021*). *Yokota et al., 2017* developed a logistic regression model to predict the risk for conversion from prediabetes to diabetes

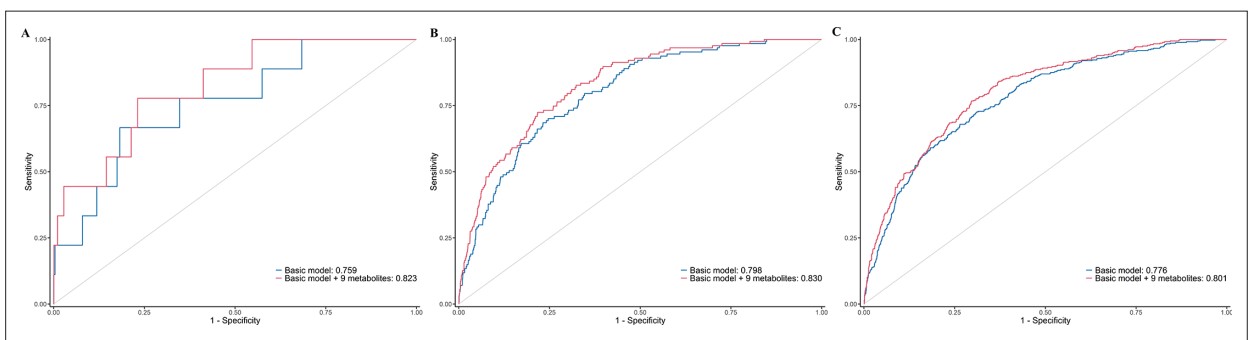

**Figure 5.** Time-dependent receiver-operating characteristic (ROC) curves of basic model and basic model plus nine metabolites for predicting 1-year (**A**), 5-year (**B**), and 10-year (**C**) risk of developing diabetes in participants with prediabetes. The basic model used conventional clinical variables including age, sex, Townsend Deprivation Index, family history of diabetes mellitus, body mass index, waist circumference, hip circumference, systolic blood pressure, diastolic blood pressure, and glycated hemoglobin A1c. The selected nine metabolites included cholesteryl esters in large HDL, triglycerides in very large VLDL, glycine, average diameter for LDL particles, tyrosine, cholesteryl esters in medium VLDL, glucose, triglycerides in IDL, and docosahexaenoic acid. HDL, high-density lipoprotein; IDL, intermediate-density lipoprotein; LDL, low-density lipoprotein; VLDL, very-low-density lipoprotein.

Medicine

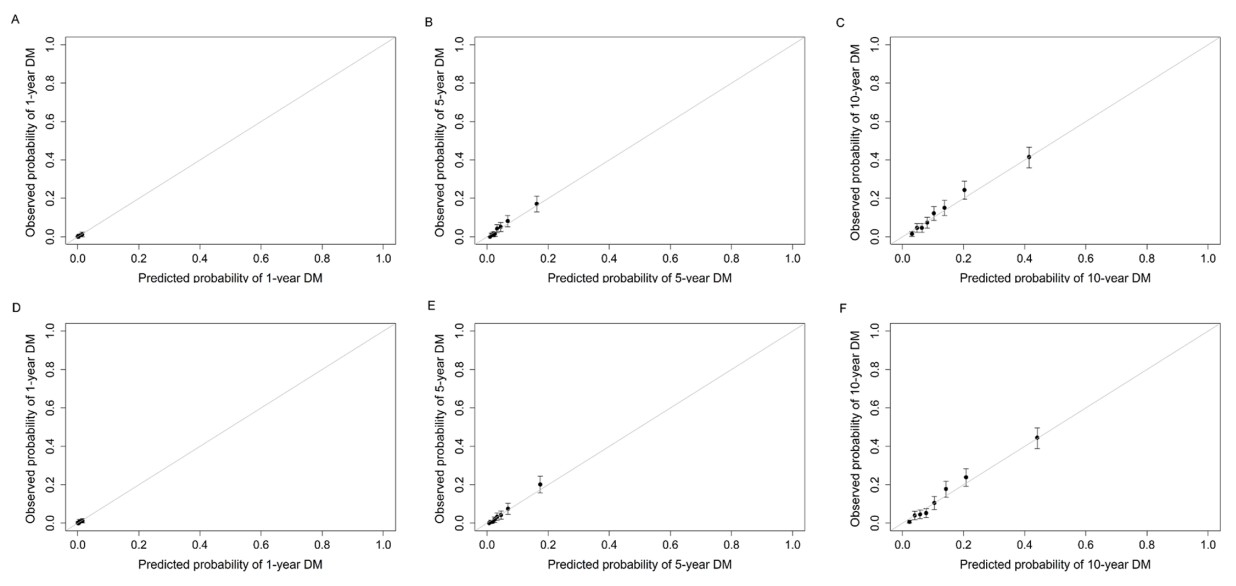

**Figure 6.** Calibration plots of basic model (**A–C**) and basic model plus nine metabolites (**D–F**) for predicting 1-year, 5-year, and 10-year risk of developing diabetes in participants with prediabetes. The basic model used conventional clinical variables including age, sex, Townsend Deprivation Index, family history of diabetes mellitus, body mass index, waist circumference, hip circumference, systolic blood pressure, diastolic blood pressure, and glycated hemoglobin A1c. The selected nine metabolites included cholesteryl esters in large HDL, triglycerides in very large VLDL, glycine, average diameter for LDL particles, tyrosine, cholesteryl esters in medium VLDL, glucose, triglycerides in IDL, and docosahexaenoic acid. HDL, high-density lipoprotein; IDL, intermediate-density lipoprotein; LDL, low-density lipoprotein; VLDL, very-low-density lipoprotein.

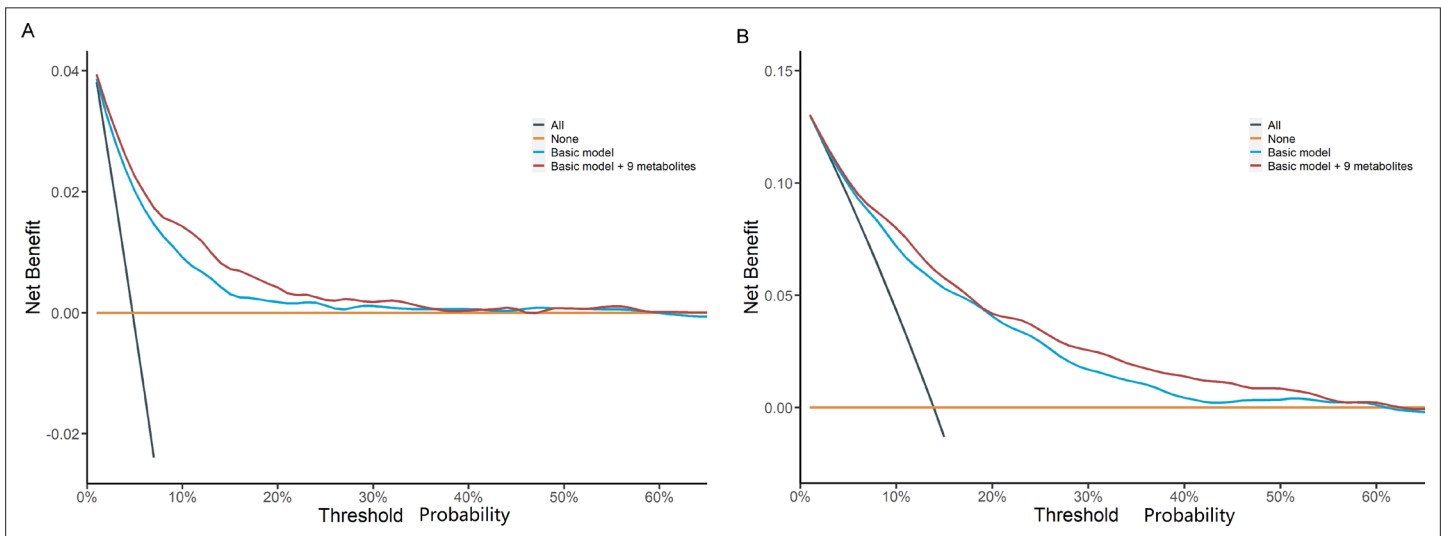

**Figure 7.** Decision curve analysis of basic model and basic model plus nine metabolites for predicting 5-year (**A**) and 10-year (**B**) risk of developing diabetes in participants with prediabetes. Decision curve analysis was not performed on 1-year prediction considering the relatively small number of prediabetic patients who develop diabetes within a year in the test set and small net benefit from intervention. The basic model used conventional clinical variables including age, sex, Townsend Deprivation Index, family history of diabetes mellitus, body mass index, waist circumference, hip circumference, systolic blood pressure, diastolic blood pressure, and glycated hemoglobin A1c. The selected nine metabolites included cholesteryl esters in large HDL, triglycerides in very large VLDL, glycine, average diameter for LDL particles, tyrosine, cholesteryl esters in medium VLDL, glucose, triglycerides in IDL, and docosahexaenoic acid. HDL, high-density lipoprotein; IDL, intermediate-density lipoprotein; LDL, low-density lipoprotein; VLDL, very-low-density lipoprotein.

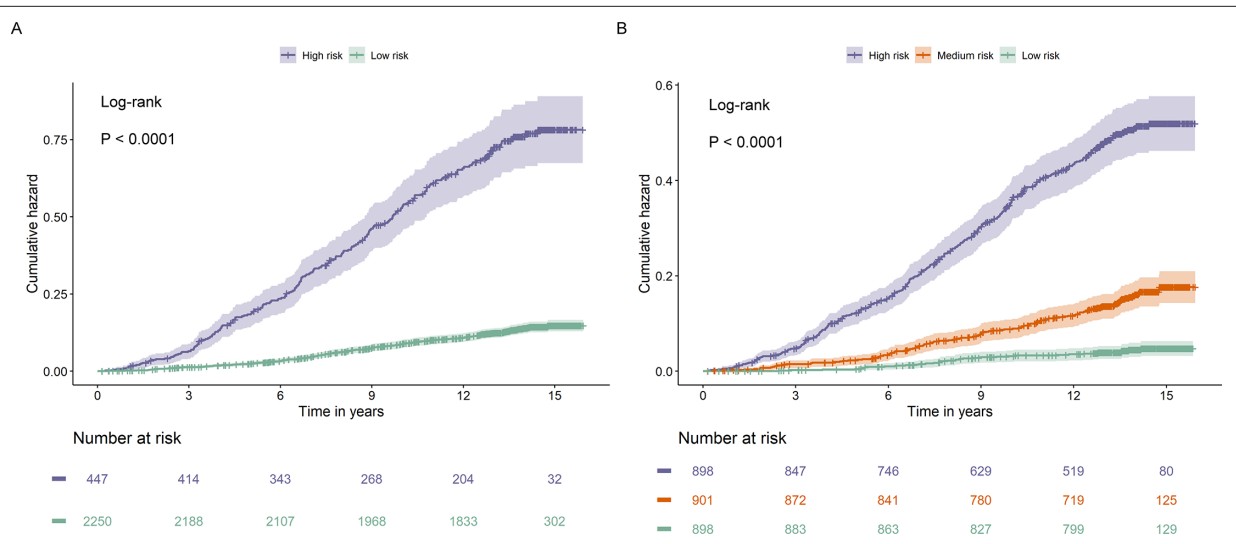

**Figure 8.** Cumulative hazard curves for participants with prediabetes with different risks stratified by the Cox model based on clinical variables and nine metabolites. The Cox model divided participants with prediabetes in the test set to two categories (**A**) and three categories (**B**) with significant differences in cumulative hazard of diabetes during the follow-up (both p < 0.0001).

The online version of this article includes the following figure supplement(s) for figure 8:

**Figure supplement 1.** Cumulative hazard curves for participants with prediabetes with different risks stratified by the Cox model based on clinical variables and nine metabolites when considering competing risk from death.

based on family history of diabetes, sex, SBP, fasting plasma glucose (FPG), HbA1c, and alanine amino-transferase. The model derived from a retrospective longitudinal study design achieved an AUROC of 0.80 (0.70–0.87) but did not take follow-up time into account. Similarly, *Liang et al., 2021* developed a predictive model using three glycemic indicators (FPG, 2 hr postprandial blood glucose [2-hPG], and HbA1c) alone and obtained a relatively low AUROC of 0.732 (95% CI 0.688–0.776). In a cohort study of 852,454 individuals with prediabetes, a machine-learning model predicting the progression to diabetes within 1 year was established using data from electronic medical records (*Cahn et al., 2020*). The model built on age, gender, BMI, medication usage, and laboratory results achieved a high AUROC of 0.865 (0.860–0.869). However, the model's performance over a longer follow-up period was unclear and conventional parameters such as lifestyle, family history of diabetes, or comorbidities were not taken into account.

Changes in circulating small-molecule metabolites may occur long before the disease onset. Although rapid development in the technology of metabolomics provides a powerful tool for precise disease prediction, few studies have investigated the role of metabolomics-derived metabolic biomarkers in predicting progression from prediabetes to diabetes. To our best knowledge, only one case–control study among 153 individuals with prediabetes and 160 matched controls reported that

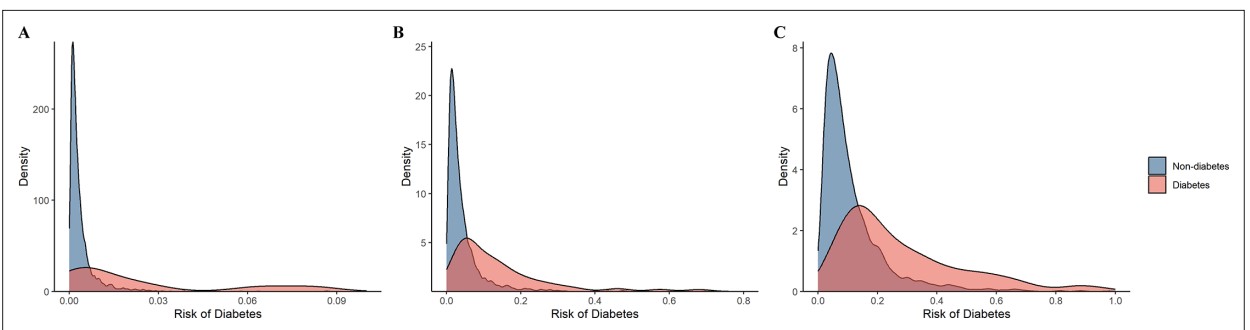

**Figure 9.** The distribution of the predictive probability of developing diabetes among participants with prediabetes by incident diabetes status within 1 year (**A**), 5 years (**B**), and 10 years (**C**).

adding 13 metabolites to conventional clinical variables including BMI, waist–hip ratio, WC, SBP, DBP, triglyceride, LDL, and triglyceride-glucose index improved the risk prediction of diabetes progression within 5 years, with the AUROC increasing from 0.72 to 0.98 (*Ren et al., 2021*). However, the predictive ability of metabolites in prospective settings with large sample size remains uncertain. In this longitudinal study among 13,489 participants with prediabetes, we comprehensively used multiple machine-learning algorithms to identify a panel of nine circulating metabolites that were associated with diabetes incidence during a median follow-up of 13.6 years. The CPH model integrating conventional clinical variables and the selected metabolic signature achieved a comparatively high AUROC of 0.823, 0.830, and 0.801 for 1-, 5-, and 10-year diabetes risk, respectively. Importantly, the addition of the metabolites resulted in a significant improvement in the discrimination ability and risk reclassification of diabetes beyond conventional risk factors. Furthermore, we categorized participants according to the optimal threshold points of the predicted value and found that the high-risk group had a significantly higher cumulative incidence of diabetes than the low-risk group. Most importantly, a model with good discrimination does not necessarily have high clinical value. Hence, DCA was used to compare the clinical utility of the model before and after adding the metabolites, and this showed a higher net benefit for the latter than the basic model, suggesting the addition of the metabolites increased the clinical value of prediction, that is, the potential benefit of guiding management in individuals with prediabetes (*Vickers et al., 2019*; *Li et al., 2023*). These results provided novel evidence supporting the value of metabolic biomarkers in risk prediction and stratification for the progression from prediabetes to diabetes. Considering the epidemic proportion of prediabetes worldwide, even a modest improvement in diabetes risk prediction among individuals with prediabetes will have substantial clinical and public health implications. Early detection of individuals with prediabetes who are at high risk of developing diabetes would not only advance targeted screening initiatives, health management, and interventions but also facilitate a rational allocation of medical resources while avoiding disproportionate healthcare expenditure, which could finally translate into precise and efficient prevention of diabetes. The value of the selected metabolic biomarkers in diabetes prediction was also confirmed in individuals with normal glucose.

Our study presents several strengths. Circulating metabolites were quantified via NMR-based metabolome profiling within the UK Biobank, which offers metabolite qualification with relatively lower costs and better reproducibility (*Geng et al., 2024*). Additional strengths of our study included large sample size, prospective study design with long-term follow-up, and comprehensive control of covariates. Moreover, we used multiple machine-learning algorithms to identify the consistently important metabolic biomarkers based on which we developed the predictive models. The final model exhibited relatively high performance for 1-, 5-, and 10-year diabetes risk prediction. However, several limitations of our study should be noted. First, since FPG and 2-hPG were not available in the UK Biobank, we defined prediabetes using HbA1c alone and to what extent our results could be extrapolated to other people with prediabetes determined by multiple glycemic indicators requires further investigation. Second, circulating metabolites were measured at baseline, thus their dynamic change over time could not be captured. However, our models showed stable performance in predicting short- and long-term progression to diabetes (1–10 years), indicating the validity of single measurements of metabolic biomarkers for risk prediction. Third, the Nightingale metabolomics platform primarily focused on lipids and lipoprotein sub-fractions, and thus the predictive value of other metabolites in the progression from prediabetes to diabetes warranted further research using an untargeted metabolomics approach. Additionally, the use of non-fasting blood samples might increase inter-individual variation in metabolic biomarker concentrations, however, fasting duration has been reported to account for only a small proportion of variation in plasma metabolic biomarker concentrations (*Li Gao et al., 2019*). Therefore, we believe the impact of non-fasting samples on our findings would be minor. Fourth, although incident diabetes cases were ascertained through different data sources, including hospital inpatient records, death registers, and primary care records, some undiagnosed diabetes might have been missed. This misclassification would underestimate the effect of the observed associations between metabolites and diabetes risk. Fifth, we could not draw any conclusion about the causality between the identified metabolites and the risk for progression to diabetes due to the observational nature, which remained to be validated in further experimental studies. Sixth, in this study, the prediction models were established and tested using the UK Biobank dataset, external validation in an independent cohort is warranted to confirm the predictive values of the metabolic biomarkers.

Finally, the participants from the UK Biobank were mostly White, which might limit the generalizability of the findings to other populations.

## Conclusions

In this large prospective study among individuals with prediabetes, we detected a panel of circulating metabolites that were associated with an increased risk of progressing to diabetes. Use of these metabolites significantly improved the risk prediction of progression from prediabetes to diabetes. Our findings provide evidence that integrating metabolite markers with conventional risk factors is a promising approach to advance effective screening strategies and precise interventions for individuals with prediabetes who are at high risk of developing diabetes.

## Acknowledgements

We thank all participants and staff in the UK Biobank for their dedication and contribution to this study.

## Additional information

### Funding

| Funder | Grant reference number | Author |
|---|---|---|
| Shanghai Municipal Health Commission | 2022XD017 | Yingli Lu |
| Innovative Research Team of High-level Local University in Shanghai | SHSMU-ZDCX20212501 | Yingli Lu |
| Shanghai Municipal Human Resources and Social Security Bureau | 2020074 | Yingli Lu |
| Clinical Research Plan of Shanghai Hospital Development Center | SHDC2020CR4006 | Yingli Lu |
| Science and Technology Commission of Shanghai Municipality | 22015810500 | Yingli Lu |

The funders had no role in study design, data collection, and interpretation, or the decision to submit the work for publication.

### Author contributions

Jiang Li, Formal analysis, Methodology, Writing – original draft; Yuefeng Yu, Formal analysis, Writing – original draft; Ying Sun, Yanqi Fu, Wenqi Shen, Lingli Cai, Data curation; Xiao Tan, Yan Cai, Ningjian Wang, Writing – review and editing; Yingli Lu, Conceptualization, Supervision, Funding acquisition; Bin Wang, Conceptualization, Supervision, Writing – review and editing

### Author ORCIDs

Jiang Li https://orcid.org/0009-0005-6253-618X
Ningjian Wang https://orcid.org/0000-0001-9591-6991
Bin Wang https://orcid.org/0000-0002-4869-1352

### Ethics

The study was approved by the Northwest Multicenter Research Ethics Committee (REC reference for UK Biobank 11/NW/0382), and all participants provided informed consent.

Reviewer #1 (Public review): https://doi.org/10.7554/eLife.98709.3.sa1
Reviewer #2 (Public review): https://doi.org/10.7554/eLife.98709.3.sa2
Author response https://doi.org/10.7554/eLife.98709.3.sa3

# Additional files

## Supplementary files

• Supplementary file 1. List of 168 NMR-based metabolomic biomarkers in the UK Biobank. HDL, high-density lipoproteins; IDL, intermediate-density lipoproteins; LDL, low-density lipoproteins; VLDL, very-low-density lipoproteins.

• Supplementary file 2. Associations of 168 metabolic biomarkers with risk of diabetes among 13,489 participants with prediabetes. Hazard ratios (HRs) were presented per 1 standard deviation (SD) higher of metabolic biomarker on the natural log scale and were adjusted for age, sex, ethnicity, education, Townsend Deprivation Index, employment status, household income, family history of diabetes, history of CVD, history of hypertension, history of dyslipidemia, history of CLD, history of cancer, body mass index, waist circumference, hip circumference, smoking status, moderate alcohol, healthy diet score, healthy sleep score, physical activity, systolic blood pressure, diastolic blood pressure and glycated hemoglobin A1c. p value <0.05/168 were highlighted in bold. Apo-A1, apolipoprotein A1; Apo-B, apolipoprotein B; Apo-LP, apolipoprotein; BMI, body mass index; CVD, cardiovascular disease; CLD, chronic lung disease; DHA, docosahexaenoic acid; FA, fatty acids; HDL, high-density lipoproteins; HDL-D, high-density lipoprotein particle diameter; IDL, intermediate-density lipoproteins; L, large; LA, linoleic acid; LDL, low-density lipoproteins; LDL-D, low-density lipoprotein particle diameter; LP, lipoprotein; M, medium; MUFA, monounsaturated fatty acids; PUFA, polyunsaturated fatty acids; S, small; SFA, saturated fatty acids; VLDL, very-low-density lipoproteins; VLDL-D, very-low-density lipoprotein particle diameter; XL, very large; XS, very small; XXL, extremely large.

• Supplementary file 3. Coefficients of the selected 17 metabolites by priority-Lasso. HDL, high-density lipoproteins; IDL, intermediate-density lipoproteins; LDL, low-density lipoproteins; VLDL, very-low-density lipoproteins.

• Supplementary file 4. Associations of the selected nine metabolites with risk of diabetes among 13,489 participants with prediabetes after adjusting for conventional clinical variables. Hazard ratios (HRs) were presented per 1 standard deviation (SD) higher of metabolic biomarker on the natural log scale and were adjusted for age, sex, Townsend Deprivation Index, family history of diabetes, body mass index, waist circumference, hip circumference, systolic blood pressure, diastolic blood pressure, and glycated hemoglobin A1c. HDL, high-density lipoproteins; IDL, intermediate-density lipoproteins; LDL, low-density lipoproteins; VLDL, very-low-density lipoproteins.

• Supplementary file 5. Performance of Cox proportional hazards prediction models for the risk of diabetes among participants with normoglycemia. (a) Basic model: age, sex, Townsend Deprivation Index, family history of diabetes mellitus, body mass index, waist circumference, hip circumference, systolic blood pressure, diastolic blood pressure, and glycated hemoglobin A1c. (b) The selected nine metabolic biomarkers: cholesteryl esters in large HDL, triglycerides in very large VLDL, glycine, average diameter for LDL particles, tyrosine, cholesteryl esters in medium VLDL, glucose, triglycerides in IDL, docosahexaenoic acid. AUROC, area under the receiver-operating characteristic curve; HDL, high-density lipoprotein; IDL, intermediate-density lipoprotein; IDI, absolute integrated discrimination improvement; LDL, low-density lipoprotein; NRI, net reclassification improvement; VLDL, very-low-density lipoprotein.

• MDAR checklist

## Data availability

The data analyzed during this study are available at https://www.ukbiobank.ac.uk/. This research has been conducted using the UK Biobank Resource under application number 77740.

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
