## [Editor Report · eLife assessment]

This **important** study combines prospective cohort, metabolomics and machine learning to identify a panel of nine circulating metabolites that improved the ability in risk prediction of progression from prediabetes to diabetes. The findings are **convincing**, and using current state-of-the-art methods the data and analyses support the claims. This paper provides insights into the integration of these metabolites into clinical and public health practice.

---

## [Referee Report · Reviewer #1 (Public review)]

Using the UK Biobank, this study assessed the value of nuclear magnetic resonance measured metabolites as predictors of progression to diabetes. The authors identified a panel of 9 circulating metabolites that improved the ability in risk prediction of progression from prediabetes to diabetes. In general, this is a well-performed study, and the findings may provide a new approach to identifying those at high risk of developing diabetes.

Comments on the revised version:

Thanks so much for carefully addressing my comments.

---

## [Referee Report · Reviewer #2 (Public review)]

Deciphering the metabolic alterations characterizing the prediabetes-diabetes spectrum could provide early time windows for targeted preventive measures to extend precision medicine while avoiding disproportionate healthcare costs. The authors identified a panel of 9 circulating metabolites combined with basic clinical variables that significantly improved the prediction from prediabetes to diabetes. These findings provided insights into the integration of these metabolites into clinical and public health practice.

Comments on the revised version:

Congratulations to the authors. I have no more comments.

---

## [Author Response]

The following is the authors’ response to the original reviews.

**Reviewer #1 (Public Review):**
Using the UK Biobank, this study assessed the value of nuclear magnetic resonance measured metabolites as predictors of progression to diabetes. The authors identified a panel of 9 circulating metabolites that improved the ability in risk prediction of progression from prediabetes to diabetes. In general, this is a well-performed study, and the findings may provide a new approach to identifying those at high risk of developing diabetes. I have some comments that may improve the importance of this study.

We deeply appreciate the reviewer's invaluable time dedicated to the review of this manuscript and the insightful comments to enhance its overall quality.

(1) It is unclear why the authors only considered the top 20 variables in the metabolite selection and why they did not set a wider threshold.

Thank you for the comment. We set the top 20 variables in the metabolite selection balancing the performance of the final diabetes risk prediction model and the clinical applicability due to measurement costs. We have added this explanation in the “Methods” section.

“We chose the intersection set of the top 20 most important variables selected by the three machine learning models, after balancing the performance of the final diabetes risk prediction model and the clinical applicability associated with measurement costs of metabolites.”

(2) The methods section would benefit from a more detailed exposition of how parameter tuning was conducted and the range of parameters explored during the training of the RSF model.

According to the reviewer’s suggestion, we have added a more detailed description of parameters tunning and the range of parameters explored during the training of the RSF model in the “Method S3” section in the Supplementary material.

“The RSF model was fitted using the “randomForestSRC” package and the grid search method was used for hyperparameter tuning. Specifically, the grid search method was used to tune hyperparameters among the RSF model, through minimizing out-of-sample or out-of-bag error1. Each tree in the RSF is constructed from a random sample of the data, typically a bootstrap sample or 63.2% of the sample size (as in the present study). Consequently, not all observations are used to construct each tree. The observations that are not used in the construction of a tree are referred to as out-of-bag observations. In an RSF model, each tree is built from a different sample of the original data, so each observation is “out-of-bag” for some of the trees. The prediction for an observation can then be obtained using only those trees for which the observation was not used for the construction. A classification for each observation is obtained in this way and the error rate can be estimated from these predictions. The resulting error rate is referred to as the out-of-bag error. Through calculating the out-of-bag error in each iteration, the best hyperparameters were finally determined.

The hyperparameters to be tuned and range of grid search in the present study were below: number of trees (50-1000, by 50), number of variables to possibly split at each node (3-6, by 1), and minimum size of terminal node (1-20, by 1)2.”

(3) It is hard to understand the meaning of the decision curve analysis and the clinical implications behind the net benefit, which are required to clarify the application values of models.

Thank you for the comment. We have added more description and discussion about the decision curve analysis in the “Methods” and “Discussion” sections.

“Furthermore, we used decision curve analysis (DCA) to assess the clinical usefulness of prediction model-based guidance for prediabetes management, which calculates a clinical “net benefit” for one or more prediction models in comparison to default strategies of treating all or no patients3.”

“Most importantly, a model with good discrimination does not necessarily have high clinical value. Hence, DCA was used to compare the clinical utility of the model before and after adding the metabolites, and this showed a higher net benefit for the latter than the basic model, suggesting the addition of the metabolites increased the clinical value of prediction, i.e., the potential benefit of guiding management in individuals with prediabetes3,4. These results provided novel evidence supporting the value of metabolic biomarkers in risk prediction and stratification for the progression from prediabetes to diabetes.”

(4) Notably, the NMR platform utilized within the UK Biobank primarily focused on lipid species. This limitation should be discussed in the manuscript to provide context for interpreting the results and acknowledge the potential bias from the measuring platform.

Thank you for the comment. We acknowledged this limitation that NMR platform within the UK Biobank primarily focused on lipid species and the potential bias from the measuring platform and have added this in “Discussion” section.

“Third, the Nightingale metabolomics platform primarily focused on lipids and lipoprotein sub-fractions, and thus the predictive value of other metabolites in the progression from prediabetes to diabetes warranted further research using an untargeted metabolomics approach.”

(5) The manuscript should explain the potential influence of non-fasting status on the findings, particularly concerning lipoprotein particles and composition. There should be a detailed discussion of how non-fasting status may impact the measurement and the findings.

According to the reviewer’s suggestion, we have added more details to explain the potential influence of non-fasting status on our findings in the “Discussion” section.

“Additionally, the use of non-fasting blood samples might increase inter-individual variation in metabolic biomarker concentrations, however, fasting duration has been reported to account for only a small proportion of variation in plasma metabolic biomarker concentrations5. Therefore, we believe the impact of non-fasting samples on our findings would be minor.”

(6) Cross-platform standardization is an issue in metabolism, and further descriptions of quality control are recommended.

Thank you for the comment. We have added more description of quality control in the “Method S1” section in the Supplementary material.

“Metabolic biomarker profiling by Nightingale Health’s NMR platform provides consistent results over time and across spectrometers. Furthermore, the sample preparation is minimal in the Nightingale Health’s metabolic biomarker platform, circumventing all extraction steps. These aspects result in highly repeatable biomarker measurements. Pre-specified quality metrics were agreed between UK Biobank and Nightingale Health to ensure consistent results across the samples, and pilot measurements were conducted. Nightingale Health performed real-time monitoring of the measurement consistency within and between spectrometers throughout the UK Biobank samples. Two control samples provided by Nightingale Health were included in each 96-well plate for tracking the consistency across multiple spectrometers. Furthermore, two blind duplicate samples provided by the UK Biobank were included in each well plate, with the position information unlocked only after results delivery. Coefficient of variation (CV) targets across the metabolic biomarker profile were pre-specified for both Nightingale Health’s internal control samples and UK Biobank’s blind duplicates. The targets were met for each consecutively measured batch of ~25,000 samples. For the majority of the metabolic biomarkers, the CVs were below 5% (https://biobank.ndph.ox.ac.uk/showcase/refer.cgi?id=3000). Further, the distributions of measured biomarkers from 5 sample batches indicated absence of batch effects (https://biobank.ctsu.ox.ac.uk/ukb/ukb/docs/nmrm_app1).”

**Reviewer #2 (Public Review):**
Deciphering the metabolic alterations characterizing the prediabetes-diabetes spectrum could provide early time windows for targeted preventive measures to extend precision medicine while avoiding disproportionate healthcare costs. The authors identified a panel of 9 circulating metabolites combined with basic clinical variables that significantly improved the prediction from prediabetes to diabetes. These findings provided insights into the integration of these metabolites into clinical and public health practice. However, the interpretation of these findings should take account of the following limitations.

We appreciate the reviewer’s positive comments and encouragement.

(1) First, the causal relationship between identified metabolites and diabetes or prediabetes deserves to be further examined particularly when the prediabetic status was partially defined. Some metabolites might be the results of prediabetes rather than the casual factors for progression to diabetes.

Thank you for your insightful comments. We agree with you that the panel of metabolites in this study might not be the causal factor for progression from prediabetes to diabetes, which needs further validation in experimental studies. We have added this limitation in the “Discussion” section.

“Fifth, we could not draw any conclusion about the causality between the identified metabolites and the risk for progression to diabetes due to the observational nature, which remained to be validated in further experimental studies.”

(2) The blood samples were taken at random (not all in a non-fasting state) and so the findings were subjected to greater variability. This should be discussed in the limitations.

According to the reviewer’s suggestion, we have added more details to explain the potential influence of non-fasting status on our findings in the “Discussion” section.

“Additionally, the use of non-fasting blood samples might increase inter-individual variation in metabolic biomarker concentrations, however, fasting duration has been reported to account for only a small proportion of variation in plasma metabolic biomarker concentrations5. Therefore, we believe the impact of non-fasting samples on our findings would be minor.”

(3) The strength of NMR in metabolic profiling compared to other techniques (i.e., mass spectrometry [MS], another commonly used metabolic profiling method) could be added in the Discussion section.

According to the reviewer’s suggestion, we have added the strength of NMR in metabolic profiling compared to other techniques in the “Discussion” section.

“Circulating metabolites were quantified via NMR-based metabolome profiling within the UK Biobank, which offers metabolite qualification with relatively lower costs and better reproducibility6.”

(4) Fourth, the applied platform focuses mostly on lipid species which may be a limitation as well.

Thank you for the comment. We acknowledged this limitation that NMR platform within the UK Biobank primarily focused on lipid species and the potential bias from the measuring platform and have added this in the “Discussion” section.

“Third, the Nightingale metabolomics platform primarily focused on lipids and lipoprotein sub-fractions, and thus the predictive value of other metabolites in the progression from prediabetes to diabetes warranted further research using an untargeted metabolomics approach.”

(5) It is a very large group with pre-diabetes, but the results only apply to prediabetes and not to the general population. This should be clear, although the authors have also validated the predictive value of these metabolites in the general population.

Thank you for the comment. We agree with you that the results only apply to prediabetes and not to the general population, though they also showed potential predictive value among participants with normoglycemia. We have accordingly modified the relevant expressions in the “Conclusion” section to restrict these findings to participants with prediabetes.

“In this large prospective study among individuals with prediabetes, we detected a panel of circulating metabolites that were associated with an increased risk of progressing to diabetes.”

**Recommendations for the Authors:**

Thank you for providing the valuable feedback and the time you have dedicated to our work.

(1) In the first paragraph of the Discussion section, please include the specific names of the metabolites selected from machine learning methods.

Thank you for your comment and we have added accordingly in the first paragraph of the “Discussion” section.

“More importantly, our findings suggested that adding the selected metabolites (i.e., cholesteryl esters in large HDL, cholesteryl esters in medium VLDL, triglycerides in very large VLDL, average diameter for LDL particles, triglycerides in IDL, glycine, tyrosine, glucose, and docosahexaenoic acid) could significantly improve the risk prediction of progression from prediabetes to diabetes beyond the conventional clinical variables.”

(2) To enhance the readability and simplicity of the paper, the description of covariate collection in the methods section should be streamlined, with detailed information provided in the supplementary materials.

Thank you for your suggestion and we have moved details about covariates collection to the “Supplementary method S2” to enhance the readability and simplicity of the paper.

“Information on covariates was collected through a self-completed touchscreen questionnaire or verbal interview at baseline, including age, sex, ethnicity, Townsend deprivation index, household income, education, employment status, smoking status, moderate alcohol, physical activity, healthy diet score, healthy sleep score, family history of diabetes, history of cardiovascular disease (CVD), history of hypertension, history of dyslipidemia, history of chronic lung diseases (CLD), and history of cancer.

Physical measurements included systolic (SBP) and diastolic blood pressure (DBP), height, weight, waist circumference (WC), and hip circumference (HC). Body mass index (BMI) was calculated as weight in kilograms divided by the square of height in meters (kg/m²). Missing covariates were imputed by the median value for continuous variables and a missing indicator for categorical variables. More details about covariates collection can be found in Method S2.”

1. Title for Table 2, using Cox proportional hazards prediction models is not common. You may consider the title "Performance of Cox proportional hazards regression models in prediction of progression of prediabetes to diabetes".

Thank you for your suggestion and we have revised it accordingly.

4. Figure 3, did the authors consider competing risk to compute cumulative incidence function?

Thank you for your comment. We did not consider competing risk from death when plotting the cumulative hazard curves. However, following your suggestion, we have included an additional cumulative hazard plot after considering the competing

References

(1) Janitza S, Hornung R. On the overestimation of random forest's out-of-bag error. PLoS One. 2018;13(8):e0201904.

(2) Tian D, Yan HJ, Huang H, et al. Machine Learning-Based Prognostic Model for Patients After Lung Transplantation. JAMA Netw Open. 2023;6(5):e2312022.

(3) Vickers AJ, van Calster B, Steyerberg EW. A simple, step-by-step guide to interpreting decision curve analysis. Diagn Progn Res. 2019;3:18.

(4) Li J, Xi F, Yu W, Sun C, Wang X. Real-Time Prediction of Sepsis in Critical Trauma Patients: Machine Learning-Based Modeling Study. JMIR Form Res. 2023;7:e42452.

(5) Li-Gao R, Hughes DA, le Cessie S, et al. Assessment of reproducibility and biological variability of fasting and postprandial plasma metabolite concentrations using 1H NMR spectroscopy. PLoS One. 2019;14(6):e0218549.

(6) Geng T-T, Chen J-X, Lu Q, et al. Nuclear Magnetic Resonance–Based Metabolomics and Risk of CKD. American Journal of Kidney Diseases. 2023.